# LoRa, Zigbee and 5G Propagation and Transmission Performance in an Indoor Environment at 868 MHz

**DOI:** 10.3390/s23063283

**Published:** 2023-03-20

**Authors:** Ricardo Robles-Enciso, Isabel Pilar Morales-Aragón, Alfredo Serna-Sabater, María Teresa Martínez-Inglés, Antonio Mateo-Aroca, Jose-María Molina-Garcia-Pardo, Leandro Juan-Llácer

**Affiliations:** 1Information Technologies and Communications Department, Universidad Politécnica de Cartagena, 30202 Cartagena, Spain; 2Departament of Automatics, Electrical Engineering and Electronic Technology, Universidad Politécnica de Cartagena, 30202 Cartagena, Spain; 3Department of Engineering and Applied Techniques, Centro Universitario de la Defensa, San Javier Air Force Base, Ministerio de Defensa-Universidad Politécnica de Cartagena, 30720 Santiago de la Ribera, Spain

**Keywords:** 5G, LoRa and Zigbee transmission, BER and RSSI, indoor corridor, log-distance model, propagation at 868 MHz, QoS link measurements

## Abstract

In this work, we present power and quality measurements of four transmissions using different emission technologies in an indoor environment, specifically a corridor, at the frequency of 868 MHz under two non-line-of-sight (NLOS) conditions. A narrowband (NB) continuous wave (CW) signal has been transmitted, and its received power has been measured with a spectrum analyzer, LoRa and Zigbee signals have also been transmitted, and their Received Signal Strength Indicator (RSSI) and bit error rate (BER) have been measured using the transceivers themselves; finally, a 20 MHz bandwidth 5G QPSK signal has also been transmitted and their quality parameters, such as SS-RSRP, SS-RSRQ and SS-RINR, have been measured using a SA. Thereafter, two fitting models, the Close-in (CI) model and the Floating-Intercept (FI) model, were used to analyze the path loss. The results show that slopes below 2 for the NLOS-1 zone and above 3 for the NLOS-2 zone have been found. Moreover, the CI and FI model behave very similarly in the NLOS-1 zone, while in the NLOS-2 zone, the CI model has poor accuracy in contrast to the FI model, which achieves the best accuracy in both NLOS situations. From these models, the power predicted with the FI model has been correlated with the measured BER value, and power margins have been established for which LoRa and Zigbee would each reach a BER greater than 5%; likewise, −18 dB has been established for the SS-RSRQ of 5G transmission.

## 1. Introduction

Currently, the interest in new wireless technologies, such as the Internet of Things (IoT), is growing exponentially, mainly due to its potential to automate processes and optimize the use of resources. The massive deployment of sensor networks is only possible if there is some way to connect the devices without the need to wire each one of them, and that would have a very high cost [1,2,3]. To overcome these limitations of wired networks, wireless transmission technologies are highly important as they provide a feasible and versatile solution. For that reason, detailed knowledge regarding the propagation characteristics of signals is essential so as to predict the losses that will be introduced by the environment [4] where the sensor network, called wireless sensor networks (WSNs), is deployed.

Today’s IoT devices are deployed using wireless communication technologies such as LoRa [5] and Zigbee [6], among others, although recently, 5G technologies have also gained prominence in IoT networks [7]. These technologies are used especially in machine-to-machine (M2M) communications; that is, they are communications not intended for humans. The use of one technology over another depends on the bandwidth, power consumption and coverage needs and the propagation characteristics of the environment. Due to the low power consumption of the devices, these networks are called Low-Power Wide-Area Networks (LPWAN) [8].

### 1.1. Technologies

LoRa is a technology mainly used for long-range communications [9,10,11] and is resistant to multipath because it makes use of the spread spectrum technique based on Chirp Spread Spectrum (CSS) modulation [12].

Zigbee is a technology that uses DS-CDMA and FDMA multiplexing [13]. It is characterized by having a mesh network topology, being very low-cost and having lower power consumption compared to LoRa. Due to its lower power consumption, it has a shorter range than LoRa [14]. The characteristics of mesh networking, short-range communication and ultra-low-power consumption make Zigbee the ideal technology for home applications [6] in wireless personal area networks (WPAN).

5G is the fifth-generation technology standard for broadband cellular networks. It is a technology that enables higher connection speeds compared to its predecessor, 4G. This technology is based on Orthogonal Frequency Division Multiplexing (OFDM) modulation, which offers high efficiency. 5G technology is characterized by providing a high connection speed, compared to traditional LTE networks, low latency and high bandwidth; it allows a large number of connected devices and achieves an efficient use of energy which increases the battery life of the devices [7].

### 1.2. Related Works

Research work has been carried out in indoor corridors at 868 MHz [15,16], where a channel attenuation model has been proposed from radio frequency (RF) measurements. Further, papers [17,18] describe measurements in the same corridor used for this work.

Other researchers have conducted their work on path loss in a LoRa transmission in various environments such as forests [19], buildings [20], long-range urban communications [21], urban areas [22], mountains (Bletterbach canyon) [23] and a block cave gold mine [24]. In [25,26,27], the empirical propagation of LoRa in indoor environments is described. Paper [28] models and discusses the behavior of different loss models of a LoRa transmission under LOS and NLOS conditions in a harsh environment. In [29], a similar procedure is performed, but in an environment with a high presence of vegetation and foliage.

On the other hand, research has been performed on: the possibility of making Zigbee-based underground miner locators that make use of the measured RSSI and a path loss model to extrapolate the distance [30]; transmission distance estimation for 2.4 GHz Zigbee applications [31]; Zigbee propagation analysis for health-care applications in hospitals [32]; ZigBee and LoRa propagation on a snowy hill area [33]; and the propagation of a Zigbee signal in an indoor environment [34].

Several studies have been proposed in the literature about the behavior of the slope, *n*, in path loss models at high frequency in indoor environments. In [35], the propagation of a millimeter-wave at 26 and 38 GHz in an indoor corridor and stairwell was studied and modeled. In the work [36], the authors studied a path loss model in an indoor commercial area at a frequency of 28 GHz and in LOS and NLOS situations. Moreover, the path loss behavior in an indoor corridor at frequencies of 14, 18 and 22 GHz has been investigated using the CI and FI model in [37]. Similarly, in [38], the impact of the antenna height on the measured path loss in an indoor corridor using the CI and FI models was analyzed. In the work [39], the design of a path loss model using CI and FI fitting at a frequency of 40 GHz in an indoor environment was studied. In the study [40], the authors present the propagation characteristics of indoor radio channels at the frequencies of 3.5, 6, 14, 23, 26 and 28 GHz, in LOS and NLOS situations; it should be noted that they perform the FI fitting of the path loss model.

In paper [41], MacCartney et al. conducted an extensive study on the propagation of millimeter-waves, 28 and 73.5 GHz, in an indoor office, presenting the results of different path loss model fitting methods (CI, FI, CIX, CIF, CIFX, ABG and ABGX models), for vertical and horizontal polarizations in LOS and NLOS situations.

Joo et al. conducted, in their work [42], a brief analysis of the behavior of path loss in urban NLOS environments, basing their measurements on a vehicle-to-vehicle (V2V) scenario using commercial vehicle-to-X (V2X) platforms.

Since the early 1990s, the behavior of UHF radio waves in different environments has been investigated. Rappaport et al. conducted research on propagation and modeling for indoor and outdoor wireless communication channels [43], characterization and fading in factory buildings [44,45], path loss prediction models for indoor multifloored buildings [46] and propagation for personal-communication systems [47], based on the use of the CI path loss model in all investigations.

Other researchers worked on the modeling of an Ultra-Wide Bandwidth Indoor Channel [48], delay spread and signal level within office buildings [49], and receiver spatial diversity propagation path-loss model in an indoor environment at 2.4 GHz [50].

Sun et al. [51] present in their paper an extensive investigation of the accuracy of three propagation models (ABG, CI and CIF models) in their use over the entire microwave and millimeter-wave (mmWave) radio spectrum in LOS and NLOS conditions. Similarly, paper [52] presents the results of the FI, CI and ABG models in an indoor environment for LOS and NLOS situations. In addition, it proposes a new model that improves the fit of the FI, CI and ABG models. Likewise, papers [53,54,55,56] present another approach to path loss model design based on the use of machine learning methods.

The results of these works have been grouped in Table 1.

### 1.3. Motivation

We note that there is extensive research on the behavior of millimeter-wave propagation in indoor environments, while for UHF radio waves, we rely mainly on the results of work performed in the 1990s and mostly with continuous wave (CW) transmissions. In addition, the research on LoRa and Zigbee propagation is sparse and not very specific on path loss models. Similarly, research on the behavior of LoRa and Zigbee BER (Bit Error Rate) with distance is scarce, and its possible correlation with a path loss model has not yet been studied. Furthermore, the relationship between the propagation of a 5G transmission with a LoRa and Zigbee transmission has also not been studied in depth.

Therefore, the main contribution of this work, motivated by the lack of similar studies in the literature, is to design a path loss propagation model for the WSN radio link for different technologies in an indoor environment, specifically a corridor, employing the license-exempt ISM (Industrial, Scientific and Medical) 868 MHz frequency band, and relate it to the BER value measured in order to be able to predict the BER with the measured power, and vice-versa. This is possible because, as discussed in [34], there is a strong relationship between the received power, Zigbee RSSI and the BER value.

To achieve this aim, it will be necessary to accurately model the relationship between power and distance for the different transmissions (narrowband, LoRa, Zigbee and 5G), so two adjustment models (CI and FI) will be proposed, and the accuracy and behavior of each one will be analyzed. After that, the power of the most accurate model will be correlated with the BER value measured at each point, and some received power margins will be established for each technology for which there will be a BER higher than 5%.

The paper is further organized as follows. Section 2 describes the environment where the measurements are carried out and the equipment that was used. Section 3 describes the methodology carried out for the measurements with each technology and the metrics used. Section 4 describes the radio channel models used. Section 5 details the results obtained from the measurements and discusses their interpretation, and finally, Section 6 contains the conclusions of this research work.

## 2. Measurement Setup Description

### 2.1. Measurement Scenario

The scenario where the set of measures has been carried out is similar to the one measured in [17,18]. This is the underground level of the Technical University of Cartagena. It mainly consists of a 50-m-long corridor, three perpendicular corridors and some laboratories used by students and researchers. The corridor walls are made of plasterboard; the floor is made of reinforced concrete, as is the ceiling. Figure 1 shows the corridor and Figure 2 shows a top view of the measurement environment, with the measurement points marked in red. The measurement points are 2 m apart. The measurement points located in the main corridor, a horizontal corridor, are what we consider to be NLOS-1 measurements, while the measurement points that continue along the perpendicular corridor, a vertical corridor and starting from point R1 are what we consider to be NLOS-2 measurements. The aim is to measure the power and BER received at each point with different technologies in both NLOS situations.

### 2.2. Measurement Equipment

A Rohde & Schwarz (Munich, Germany) SMB100A signal generator was used to generate a narrowband transmission signal. A Rohde & Schwarz CMW500 signal generator was used to generate a 5G signal. The received power of the narrowband and 5G signals was measured using a spectrum analyzer (SA) model Anritsu (Atsugi, Kanagawa Prefecture, Japan) MS2090A. For the LoRa and the Zigbee transmissions, a LoRa PyCom (Eindhoven, NB, The Netherlands) LoPy${}^{4}$ module and a Zigbee XBee (Hopkins, MN, USA) 868LP module were used, respectively. All transmissions were made with vertical polarization and λ/4 monopole type antennas with resonance at the frequency of 868 MHz.

## 3. Methodology

### 3.1. Ideal Narrowband Measurements at 868 MHz

The study of the power attenuation of narrowband (NB) transmission is carried out following the scheme shown in Figure 3. This procedure consisted of generating a sinusoidal signal with the Rohde & Schwarz SMB100A signal generator, with a power of 0 dBm, and then amplifying it with a Mini Circuits (Brooklyn, NY, USA) ZHL-42 amplifier to have 30 dBm of transmission power before transmitting it with the antenna. The measurement of the received power at each measurement point was performed using the SA Anritsu MS2090A and using an antenna identical to the transmission one.

Due to the fact that the measurements carried out with the SA vary rapidly between values, it was decided to configure a mean filter in the SA with a window of 100 samples. This filter was intended to obtain a static measurement value at each measurement point. The specific configuration used in the SA is listed in Table 2.

### 3.2. LoRa Measurement Descriptions

The study of the signal quality and power when using LoRa technology was performed from two Pycom Lopy${}^{4}$ modules, which allow point-to-point communication to be established using LoRa frequencies. These modules allow working at different frequencies, but, in this case, a frequency of 868 MHz was used. In addition, this type of microcontroller allows wireless communications between both nodes and processing of the transmitted data. Figure 4 shows the block diagram that represents the measurement configuration covered by this technology.

The configuration of each module as a transmitter or receiver was carried out through the functions that are implemented in the modules themselves.

#### 3.2.1. Pycom Transmitter Module

This module was programmed with the aim of presenting two functions:Generate a known data string.Send this chain to another Pycom module wirelessly and at a certain speed.

Through the implemented program, the user decides the data-sending speed and once that speed has been assigned, the module sends known characters every certain period of time. In this study, the sending speed was 2 data/s.

#### 3.2.2. Receiver Pycom Module

Like the previous module, this node was also programmed to perform two functions:Receive the characters sent by the other module.Carry out an analysis of the data obtained in order to be able to estimate both the signal quality rate (BER) and the signal power (RSSI) at each point.

In this module, a program was implemented that was capable of receiving data at the maximum possible speed. Furthermore, since the characters sent were known, the module was capable of detecting those characters that were wrong or that had not been received directly during a given sampling time. In this way, it was possible to estimate the quality rate of the signal generated by the emitter module. In addition, the value of the signal power (RSSI) with which data was received at each of the study points was captured.

### 3.3. Zigbee Measurement Descriptions

To study the quality of the signal presented by communication using Zigbee technology, two XBee 868LP modules were used that work at a frequency of 868 MHz. Both modules were configured to exchange information through point-to-point communication in transparent mode. This type of communication gives the same result as if the two modules were connected by a cable, but wireless communication makes that physical cable unnecessary. XBee modules can be connected to a microcontroller via serial communication, thus allowing the information received by the module to be processed and, therefore, to monitor or even control remote devices by sending messages through their local XBee module. In this case, a Pycom Lopy${}^{4}$ microcontroller was used for each Xbee module. In this way, one of the microcontrollers sends data via serial communication to its local XBee module so that it can later be wirelessly transmitted to the other XBee device and processed by the other microcontroller. This device configuration can be seen in the block diagram shown in Figure 5.

In a similar way to communication by LoRa technology, a program was generated for Zigbee that would allow the data sent and received to be processed.

#### 3.3.1. XBee Transmitter Module

One of the XBee modules was configured in order to establish transparent communication in sender mode with the other node. In this way, this module only presented the function of sending characters every certain period of time. In addition, it should be noted that both the characters and the sending speed are configured by the Pycom module, which is connected by serial communication. Finally, it is worth mentioning that a program similar to the one used with LoRa was implemented in the Pycom module. In this way, this microcontroller allows the user to choose the sending speed, as well as to send, by serial communication to its local XBee module, known characters every certain period of time. In this study, the sending speed was the same as with the LoRa technology, 2 data/s.

#### 3.3.2. Receiver XBee Module

This module was configured with the aim of establishing wireless communication with the sending node and, therefore, acting as a receiver at the maximum speed allowed by the module. The received data were analyzed through another Pycom module which obtained, by serial communication, the data received by the XBee receiver module. In this way, as with LoRa, the characters sent were known and, therefore, knowing which data were badly received for a given sampling time, it was possible to estimate the quality rate of the signal generated by the emitting XBee module. Likewise, the value of the signal power (RSSI) with which the data was received was captured.

### 3.4. 5G QPSK at 868 MHz Measurement Descriptions

Figure 6 shows the configuration for 5G technology where a 5G NR signal was generated using a Wideband Radio Communication Tester (Rohde & Schwarz CMW500) and demodulated with a SA (Anritsu MS2090A). After programming the NR signal in MATLAB (5G Toolbox), the ARB Toolbox R&S was used to convert the .mat file to a .wv file that carries baseband signal digital IQ data. The CMW500’s ARB Generator (using the General Purpose Radio Frequency Generator Application) can generate a modulated RF signal from a waveform file. The generated signal has a subcarrier spacing of 15 kHz, 20 MHz bandwidth, QPSK modulation, 100 subframes, a normal cyclic prefix of 4.7 μs duration and hence 14 OFDM symbols (66.7 μs duration) per slot. Frequency sweep is performed, and for each frequency a power sweep is carried out; this process is automated through MATLAB’s scripts using SCPI commands.

Through the Anritsu MS2090A license 0888 NR SA application is enabled, offering several values related to signal quality measurement. The center frequency, channel bandwidth and sub-carrier spacing of the new radio signal were selected for each measurement case. SCPI commands were used to automate the measurement process and for obtaining Synchronization Signal Block (SSB) data such as the Synchronization Signal-Reference Signal Received Quality (SS-RSRQ) and Synchronization Signal-Reference Signal Received Power (SS-RSRP). The data are sent in the JSON file type.

### 3.5. Distance Metric

The distance associated with each NLOS-1 zone measurement is defined as the distance in meters between the first measured point, the closest to Tx, and the new measurement point. For measurements in the NLOS-2 zone, the distance is defined as the sum of the total NLOS-1 zone distance, i.e., the distance between the first measurement point and point R1, plus the distance between point R1 and the measurement point, a similar approach to that adopted in [57].

### 3.6. Power Metric

The power parameter used for narrowband transmission is the power measured by the SA. For 5G transmission, the SS-RSRP parameter measured by the SA is used as the power parameter. For LoRa and Zigbee transmissions, the RSSI measured by the transceiver modules is used. In order to make the ranges of each parameter comparable, a calibration will be made for each one, which will be explained in later sections.

### 3.7. Quality Metric

The quality parameter used for LoRa and Zigbee transmissions is the BER measured by the transceiver modules. BER is a parameter that represents the number of bit errors divided by the total number of bits received in a transmission. For 5G transmissions, the SS-RSRQ parameter measured by the SA is used as the quality parameter because it is used in 5G NR networks to determine the radio channel quality.

### 3.8. Measured Path Loss

The measured path loss values can be estimated by knowing the transmitted power, the gain of the transmitting and receiving antennas and the received power at the Rx according to Equation (Equation 1) [37].
(1)$$PL\left(d\right)\left[dB\right]=P_{t}-P_{r}\left(d\right)+G_{t}+G_{r}$$
where $P_{t}$ is the transmitted power in dBm, $P_{r}$ is the received power in dBm and $G_{t}$ and $G_{r}$ are the gain of the transmitting and receiving antennas in dBi. The power, $P_{r}$, will be the one measured with the different measuring instruments used in this work.

## 4. Radio Channel Models

In this section, we introduce a simple but effective propagation model based on power measurements, designed following the widely used methodology in indoor modeling. This model consists of making a linear regression, using the least squares error method, of the power values expressed on a logarithmic scale with the value of the distance from the measured point. In this way, a straight line is achieved with a specific slope that relates the logarithmic power to the distance. This slope expresses the physical way in which the transmission power decays logarithmically with distance. This modeling method is known as the Log-distance Path Loss Model [58], and it is used to derive the path loss exponent, also known as the decay factor, which is specific for each environment and frequency. This model is also known as the Close-In free space reference distance (CI) path loss fitting model [57]. The equation of this model is as follows:(2)$$PL^{CI}\left(d\right)\left[dB\right]=L_{0}+10n\log_{10}\left(\frac{d}{d_{0}}\right)+X_{\sigma }^{CI}$$
where $L_{0}$ is the free space path loss at 1 m in dB according to Equation (Equation 3), *n* is the decay factor, *d* is the distance expressed in meters, $d_{0}=1$ m, and $X_{\sigma }^{CI}$ is a lognormal variable with the standard deviation of σ in dB.
(3)$$L_{0}\left(dB\right)=10\log_{10}{\left(\frac{4\pi f_{c}}{c}\right)}^{2}$$
where $f_{c}$ is the frequency used ($868\;\times \;10^{6}$ Hz) and *c* is the speed of light (using $3\;\times \;10^{8}$ m/s).

This fitting model represents the physical behavior of the propagation attenuation by means of the parameter *n*, but it has the disadvantage of not being suitable for measurement fits under NLOS conditions [57], so an alternative model is proposed for greater precision in this circumstance.

The proposed alternative option is the Floating-Intercept (FI) path loss fitting model used in the WINNER II and 3GPP channel models [59]. In order to know the correct slope of the NLOS zone, the FI model uses two parameters, α and β, to provide the best minimum error fit. This model is given in Equation (Equation 4). The modeling will be applied by separating the NLOS-1 zone measurements from those of the NLOS-2 zone in order to know the specific path loss exponents for these two scenarios
(4)$$PL^{FI}\left(d\right)\left[dB\right]=\alpha +10\beta \log_{10}\left(d\right)+X_{\sigma }^{FI}$$
where α is a Floating-Intercept coefficient in dB, β is a distance-dependent coefficient, and $X_{\sigma }^{FI}$ is a lognormal variable with the standard deviation of σ in dB.

## 5. Measurement Results and Discussion

This section presents the results of the power and quality measurements performed for each technology. Based on these data, the design of the path loss models and the relationship of the BER with the designed models will be discussed.

### 5.1. Power Measurements and Path Loss Modeling

#### 5.1.1. Narrowband at 868 MHz

For the narrowband measurements in the NLOS-1 zone, a fit is made, and a slope of n=1.97 for the CI model and β=1.94 for the FI model is obtained, while for the measurements in the NLOS-2 zone, the slope is n=3.03 for the CI model and β=7.26 for the FI model. The measured path loss and the models are shown in Figure 7, together with the dispersion and cumulative distribution function (CDF) of the models for both NLOS zones. The parameters of the path loss models are summarized in Table 3. Analyzing the slopes, we observe that for the NLOS-1 zone, the slope value is in agreement with the result of the work [16] performed in a similar environment and at the same frequency. Meanwhile, in the NLOS-2 zone, the value of the slope increases considerably, which is due to the fact that the signal loses a large part of its energy in diffraction at point R1, corresponding to the wedge between corridors.

#### 5.1.2. LoRa at 868 MHz

Due to the fact that RSSI is a power parameter that does not represent the real received power but a relative measure with a range of values mostly defined by each chip vendor [60], the RSSI measurements taken are calibrated to refer to the standardized anchor point of a 1 m close-in free space reference distance [41,59], where $d_{0}=1$ m is the standard for an indoor system [43].

Following a procedure similar to the previous one but applied to the LoRa RSSI measurements, slopes of n=1.96 for the CI model and β=1.72 for the FI model in the NLOS-1 zone are obtained. For the NLOS-2 zone, slopes of n=3.01 for the CI model and β=4.59 for the FI model are obtained. Figure 8 shows the measurements and the models, together with their dispersion and CDF; and the parameters of the path loss models are summarized in Table 3. It is observed that the FI slope in the NLOS-1 zone (β=1.72) is very similar to the FI slope in the NLOS-1 zone for narrowband transmission (β=1.94). In the NLOS-2 zone, the FI slope (β=4.59) grows less than the FI slope of the narrowband transmission (β=7.26), mainly due to resistance to multipath fading of LoRa CSS modulation [61]. The results of the CI model are very similar to those of the other works [26,27].

#### 5.1.3. Zigbee at 868 MHz

Following the same procedure, the RSSI values of the Zigbee measurements, path loss models, their dispersion and CDF are shown in Figure 9, and the parameters of the propagation models are summarized in Table 3. Using this technology, it has only been possible to make measurements in the NLOS-1 zone, resulting in a CI slope of n=1.98 and FI slope of β=1.69, due to the fact that the signal power has been attenuated enough to not overcome the diffraction at the R1 point.

#### 5.1.4. 5G QPSK at 868 MHz

Three link quality parameters, SS-RSRP, SS-RSRQ and SS-RINR, are used in 5G measurements. For each quality parameter, four values representing each transmitted beam are obtained. As the transmission is performed by a monopole antenna, the four beams are transmitted with the same antenna. Therefore, they will have the same value since they do not travel along different paths. This can be seen in Figure 10, where the value of all the measurements made is represented in the upper part, while the lower part represents the dispersion in the quality parameters of each beam with respect to beam 0. Therefore, it can be concluded that they have the same value, and we proceed to use, from now on, only the values of the quality parameters of beam 0. The quality parameter measurements and their regression can be seen in Figure 11. Using the SS-RSRP parameter, which is equivalent to the received power [62], the CI and FI models are performed, and a slope value of n=1.91 is obtained for the CI model and a slope value of β=1.11 for the FI model, both in the NLOS-1 zone. For the NLOS-2 zone, a slope value of n=3.15 is obtained for the CI model and a slope value of β=7.39 for the FI model. Figure 12 shows the measurement values in the CI and FI models, their dispersion and CDF. The parameters of the path loss model are summarized in Table 3.

#### 5.1.5. Discussion of Path Loss Models

In the following discussion, the slope parameter of the path loss model of each technology will be analyzed. Table 1 shows the *n* value of the model from other works carried out in different environments under LOS and NLOS conditions. Table 3 groups the parameters of the models designed in this work.

For free space propagation, the slope value of the CI model is equal to 2, and when the slope value is less than 2 it could be attributed to a waveguide effect [37,41] due to the physical form of the corridor [16]. This only occurs in the CI model since it is the one that contemplates the physical behavior of the propagation by means of the *n* parameter; however, physical behavior is absent in the FI model [63].

As mentioned in Section 4, the CI model is not suitable for NLOS conditions because it does not achieve the best minimum error fit [57]. Nevertheless, it is suitable for some NLOS conditions, which can be verified by observing that the NLOS-1 slope values in the CI and FI models of the narrowband transmission are very similar, with an *n* of 1.97 and β of 1.94, respectively, because both models have the same NLOS behavior in this environment [37,59]. However, the NLOS-2 slope values in the CI and FI models are not similar, *n* of 3.03 and β of 7.26, respectively, because the CI model adjusts with respect to $L_{0}$ in the NLOS-1 zone, while the FI model makes use of the free parameter α for the precise adjustment in the NLOS-2 zone, being independent of the free space propagation behavior in the NLOS-1 zone [57], thus achieving a precise fit, but detached from the physical behavior. Therefore, the FI model does not allow insightful information to be extracted immediately from the alpha and beta parameters [59].

For LoRa, Zigbee and 5G transmissions, the CI model shows a slope value between 1.91 and 1.98 in the NLOS-1 zone, which is near to those expected in indoor LOS conditions (see Table 1). On the other hand, for the NLOS-2 zone, the slope value of the CI model is between 3 and 3.15 for NB, LoRa and 5G transmissions, also very close to the values of other studies (see Table 1), where, fundamentally, the values of *n* for the LOS zone are around 2 and the values of *n* for the NLOS zone are around 3 for an indoor environment that is mainly diaphanous and somewhat narrow, like the corridor studied in this work. Therefore, it can be observed that the NLOS-1 zone behaves very similarly to a LOS condition.

The CI model is widely used by other researchers because the use of $d_{0}=1$ m provides an easy comparison of various measurement campaigns in different environments [59].

The worst fit (σ) of the CI model with respect to the FI model may be due to the fact that the power measurements taken in LoRa, Zigbee and 5G transmissions make use of their own power indicators, RSSI and SS-RSRP, which operate in their own range and have no direct relationship with the $L_{0}$ factor resulting from the calculation of free space losses for a distance of one meter. Therefore, the fit of the CI model will be partially erroneous in these cases, although a large difference has not been found in this work.

From the data in Table 1 and Table 3, we can outline the following conclusions:The results of the CI and FI models in the NLOS-1 zone exhibit behavior very similar to the results of other studies under LOS conditions.The CI model provides a correct fit for LOS measurements but is not suitable for NLOS measurements. This model considers the physical behavior of the propagation using the *n* parameter since it adjusts the path loss to a reference distance $d_{0}$. Because it is fitted by a reference parameter, it is possible to compare the results of different measurement campaigns immediately.The FI model provides the best fit for both LOS and NLOS situations because the fit is made by two free parameters, α and β, although it is difficult to compare results and draw immediate conclusions.The CI and FI models give the same result when $L_{0}$ and α are equal [38]. This can only occur in the LOS situation, as this is when both models could have the same behavior.The CI and FI models are practically very similar in LOS measurements; in NLOS measurements, they vary significantly from each other.For narrowband transmission, the slope in the NLOS-1 zone is very close to the free space propagation value with a small waveguide effect. For the NLOS-2 zone, the slope value increases considerably due to a large energy loss in diffraction at the R1 point.For narrowband transmission and LoRa RSSI, in general, the power measurements performed have a very similar behavior between one technology and the other, with the nuance that LoRa has somewhat lower losses in the NLOS-2 zone. As shown in Figure 13, the received power curve of the narrowband transmission is very similar to that of the LoRa RSSI, previously raised by means of an offset. This offset is calculated as the difference in the mean of the narrowband transmission power measurements with the mean of the LoRa RSSI measurements. The error is calculated as the absolute value of the difference in each point of the narrowband transmission power measurement with the LoRa RSSI measurement after shifting the LoRa RSSI point cloud by adding the offset.For the RSSI of LoRa and Zigbee, continuing with the previous point, we can observe how the behavior of the RSSI measurements of both LoRa and Zigbee is very similar to the power measurements of narrowband transmission, with the slight differences indicated above, so we can conclude that the RSSI measurements are reliable to make a propagation loss model.For LoRa RSSI, the slope in the NLOS-1 zone is less close to the free space propagation, so some waveguide effect is present. In the NLOS-2 zone, the slope increases due to diffraction losses at the R1 point, but it is observed that the β value is considerably lower than in narrowband transmission due to the robustness of LoRa CSS modulation.For Zigbee RSSI, the NLOS-1 slope is almost identical to the LoRa transmission because they experience the same waveguide effects. In the NLOS-2 zone, it was not possible to measure because the signal attenuated before reaching the R1 point due to the low power at which Zigbee transmits and its lower receive sensitivity.For 5G SS-RSRP, it is the transmission that attenuates less with distance and suffers a very noticeable waveguide effect, seeing that its slope value is very close to 1 in the NLOS-1 zone for the FI model. On the other hand, in the NLOS-2 zone, it has a higher β slope than LoRa transmission and a very similar to narrowband transmission. Therefore, it is observed that 5G transmission shows the least slope in the NLOS-1 zone, while in the NLOS-2 zone, it shows a higher slope.

### 5.2. BER Measurements and Relation with the FI Path Loss Model

In this section, the BER measured under each technology is compared with the previously designed FI path loss models. Hence, the received power estimated by each model is related to the empirical quality parameter measured at each point in order to predict the BER with a measure of power or RSSI. The FI model has been chosen because it is the best-fitting model.

Figure 11 shows the SS-RSRQ parameter, 5G quality parameter, with distance. Figure 14 shows the BER value with distance in LoRa and Zigbee transmissions together with the SS-RSRQ parameter. It is observed how the transmission with Zigbee technology reaches a BER of 100% at 40 m in the NLOS-1 zone, while LoRa transmission has a BER of 0% at the same point. This is due, as already discussed in previous sections, to the fact that the transmission range of Zigbee technology is much lower than LoRa technology because Zigbee transmits with less power and has less sensitivity than LoRa. For this reason, Zigbee is limited to middle-range communications, while LoRa is for long-range communications.

#### 5.2.1. LoRa BER

Each measured LoRa BER value, previously shown in Figure 14, is grouped with the power estimation made by each loss model at each particular point, including the NLOS-1 and NLOS-2 zones. Thus, four vectors are obtained where the abscissa axis is the received power predicted by the model for a specific point, and the ordinate axis is the empirical LoRa BER measured at that point. This relationship between the predicted power for each model and the measured LoRa BER is shown in Figure 15. From this figure, we can extract that the LoRa BER does not increase to a value greater than 5% until a measured power of −88 dBm or a LoRa RSSI of −119 dBm is reached, while for Zigbee RSSI and SS-RSRP, it remains below 5% over the entire measurement range. The first row of Table 4 shows these results.

#### 5.2.2. Zigbee BER

Following a similar procedure to the previous one but with the BER value measured with Zigbee, four vectors can be composed in the same way, where the abscissa axis is the received power predicted by the model for a specific point, and the ordinate axis is the empirical Zigbee BER measured at that point. This relationship between the predicted power for each model and the measured Zigbee BER is shown in Figure 16. It is observed that to have a Zigbee BER higher than 5%, it is necessary to have a power less than −56 dBm in narrowband transmission, −94 dBm of LoRa RSSI, −97 dBm of Zigbee RSSI and −98 of SS-RSRP. The second row of Table 4 shows these results.

#### 5.2.3. 5G QPSK SS-RSRQ

For 5G transmission, SS-RSRQ was used as the quality parameter instead of BER. Relating this quality parameter at each point with the received power prediction of each model, we obtain the results shown in Figure 17. From this figure, it can be seen that to have a value of SS-RSRQ less than −18 dB requires a lower power value of −81 dBm for narrowband transmission and −114 dBm RSSI for LoRa transmission. The third row of Table 4 shows these results.

Using Figure 15, Figure 16 and Figure 17, an estimation can be made of the possible BER and SS-RSRQ that a LoRa, Zigbee and/or 5G transmission may have by means of a received power measurement of a narrowband, LoRa, Zigbee and/or 5G one. Table 5 shows, as an example, the value of arbitrary powers received in each technology and their respective BER and SS-RSRQ estimation.

Hence, we can conclude that it is possible to predict the BER of a LoRa and Zigbee transmission, and the SS-RSRQ of a 5G transmission, using only the received power in any of the technologies used in this work. Similarly, it is possible to predict the received power value in each technology using the BER value of LoRa, the BER value of Zigbee or the SS-RSRQ value of 5G.

## 6. Conclusions

This work was focused on the study of the propagation and performance aspect of NB, LoRa, Zigbee and 5G transmissions. For this purpose, indoor measurements were performed with four technologies in two NLOS situations at 868 MHz.

Two different fitting models have been proposed for the measurements, the CI and the FI models. It has been observed that the CI model shows a behavior more associated with the physical propagation of the wave itself, while the FI model shows a better fit. It has also been shown that the CI model has disadvantages in the NLOS situation, while the FI model does not. The behavior of the RSSI parameter has been found to be very similar to that of the actual measured power. The four transmissions show very similar waveguide behavior in the NLOS-1 zone (n<2), while in the NLOS-2 zone, the transmissions attenuate more aggressively, with LoRa showing less attenuation in the NLOS-2 zone (β=4.59). This indicates that LoRa has a higher performance than the other technologies used in this work for indoor corridor environments.

It has been observed that the BER of LoRa and Zigbee transmissions are strongly related to the distance, so it is also strongly related to the received power. That is to say, the BER is directly proportional to the distance and inversely proportional to the received power. By means of the FI models designed for each technology, the predicted power at a point has been correlated with the measured BER value, and power margins have been established for which a BER of 5% is exceeded. Some graphs have also been presented with which the values of the quality parameters, BER and SS-RSRQ, can be obtained by means of the measured power value and vice versa.

Therefore, it is possible to predict, to some extent, quality parameters, such as BER and SS-RSRQ, by using measured power parameters, such as real power, RSSI power and SS-RSRP power.

## Figures and Tables

**Figure 1 sensors-23-03283-f001:**
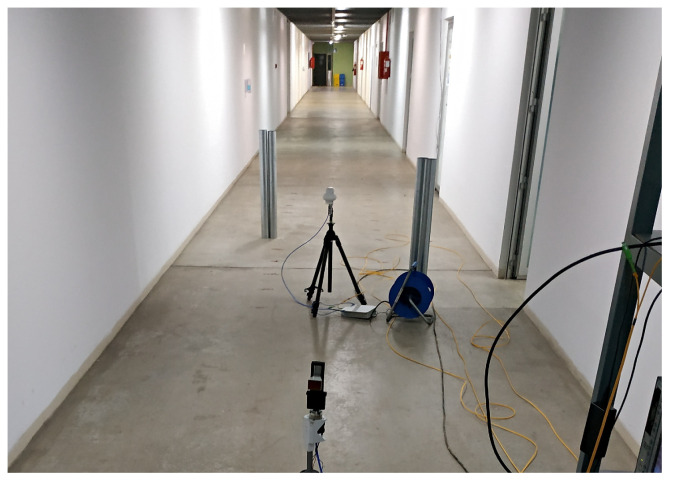
File photo of the corridor.

**Figure 2 sensors-23-03283-f002:**
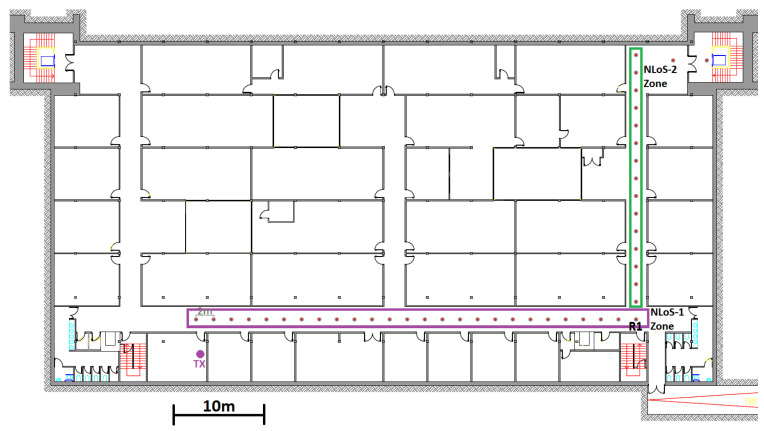
Floor plan underground level.

**Figure 3 sensors-23-03283-f003:**
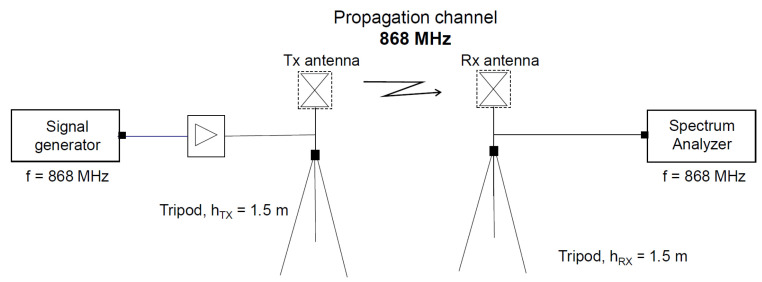
Block diagram of narrowband transmission.

**Figure 4 sensors-23-03283-f004:**
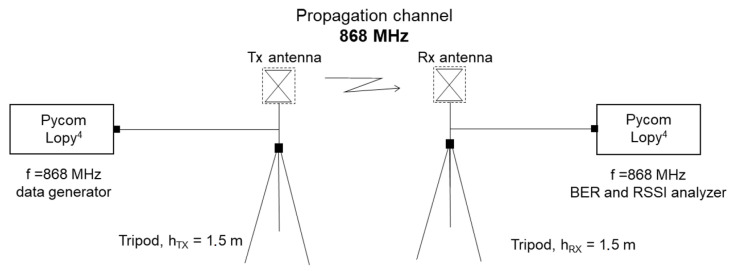
Measurement configuration with LoRa technology.

**Figure 5 sensors-23-03283-f005:**
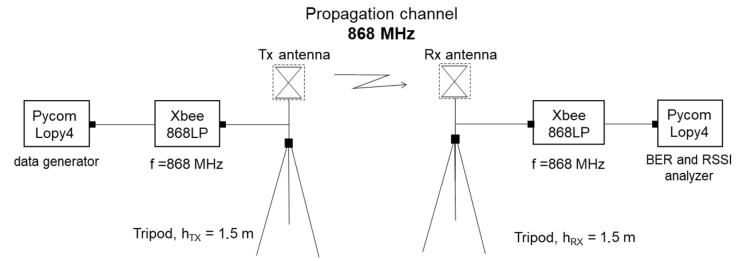
Measurement configuration with Zigbee technology.

**Figure 6 sensors-23-03283-f006:**
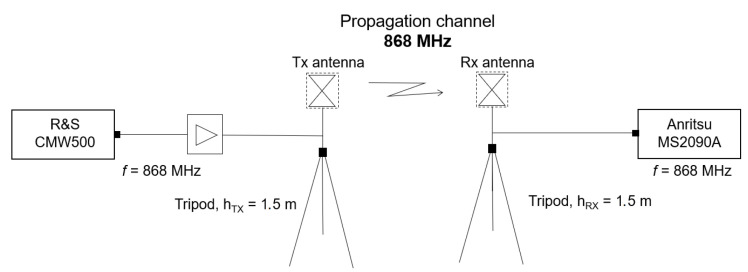
Measurement configuration with 5G technology.

**Figure 7 sensors-23-03283-f007:**
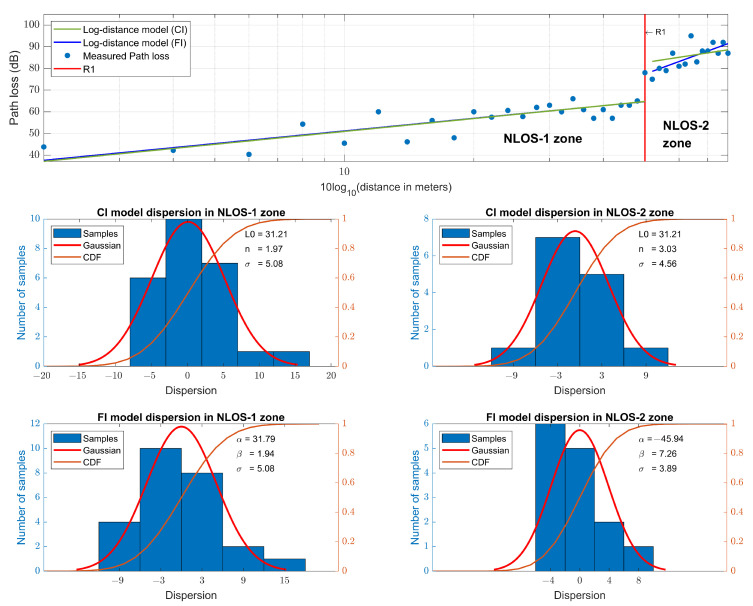
Narrowband path loss models at 868 MHz and their dispersion.

**Figure 8 sensors-23-03283-f008:**
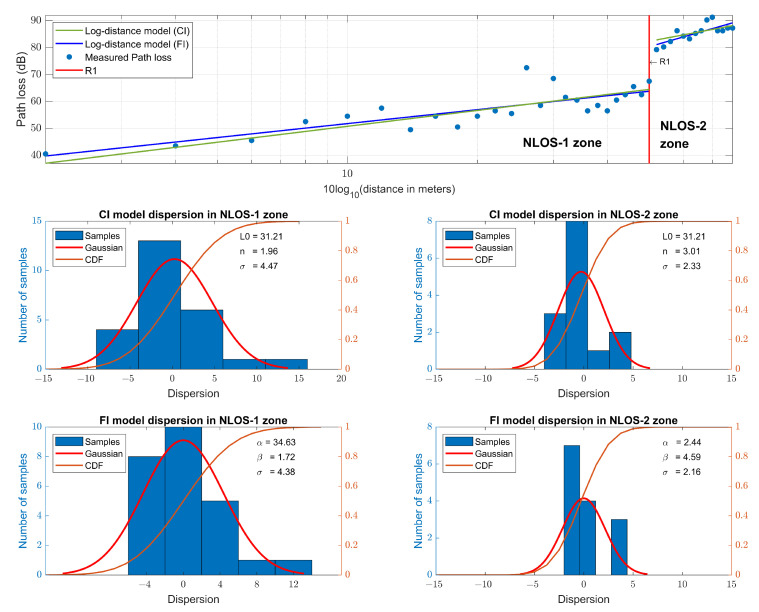
LoRa RSSI path loss models at 868 MHz and their dispersion.

**Figure 9 sensors-23-03283-f009:**
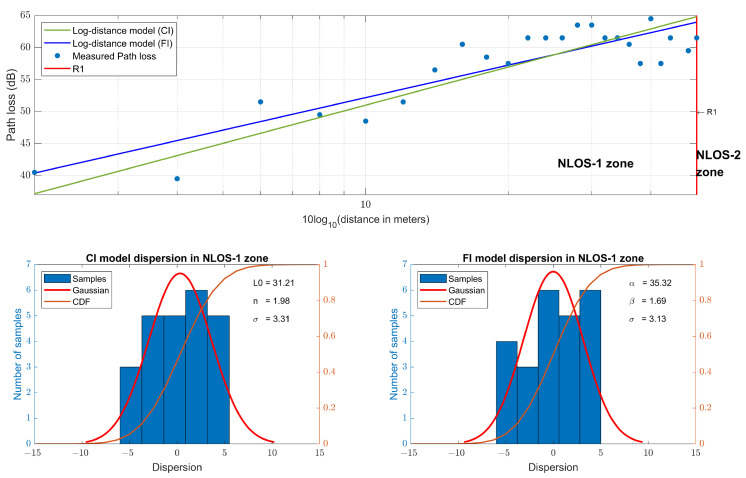
Zigbee RSSI path loss models at 868 MHz and their dispersion.

**Figure 10 sensors-23-03283-f010:**
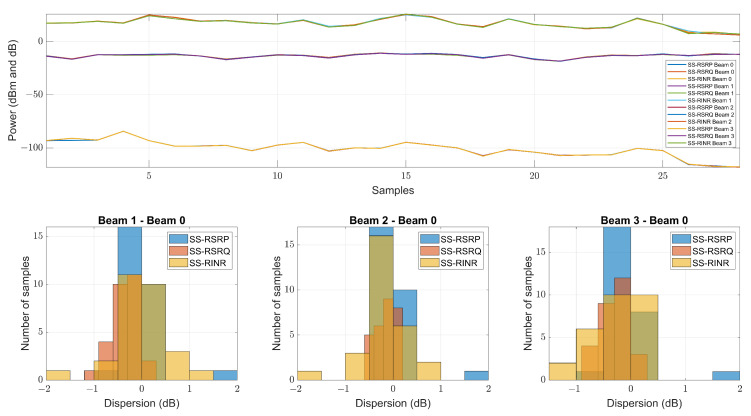
Experimental data of 5G measurements at 868 MHz.

**Figure 11 sensors-23-03283-f011:**
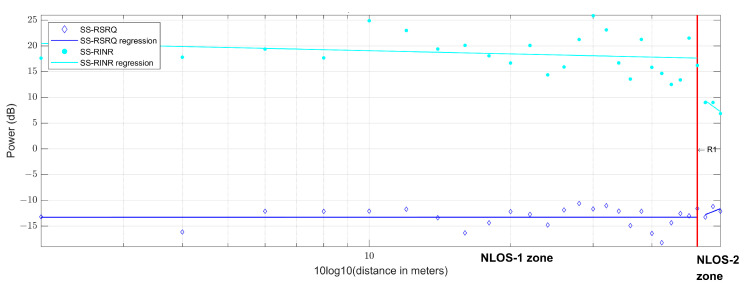
Experimental data of 5G SS-RSRQ and SS-RINR at 868 MHz.

**Figure 12 sensors-23-03283-f012:**
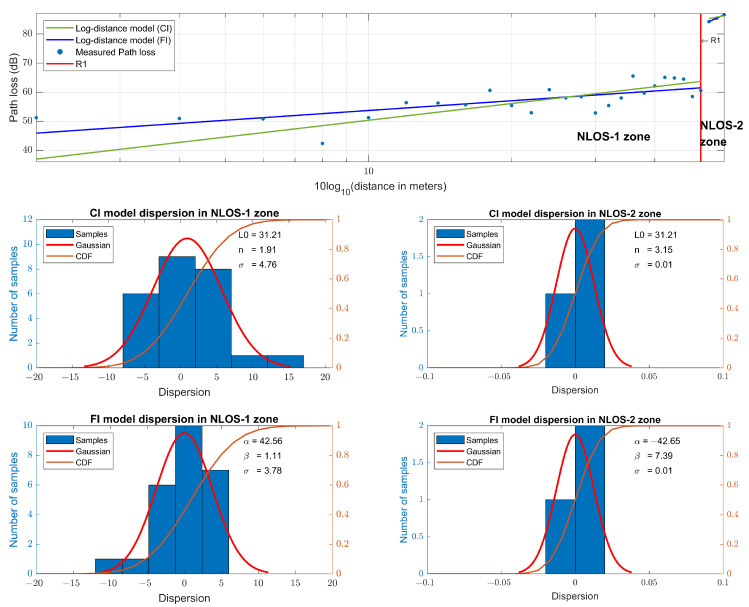
5G SS-RSRP path loss models at 868 MHz and their dispersion.

**Figure 13 sensors-23-03283-f013:**
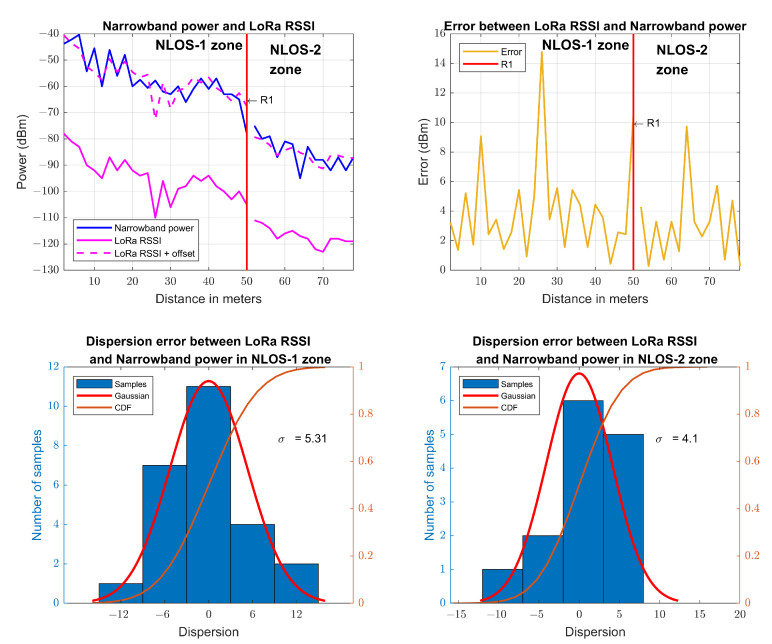
Comparison of narrowband transmission received power and LoRa RSSI.

**Figure 14 sensors-23-03283-f014:**
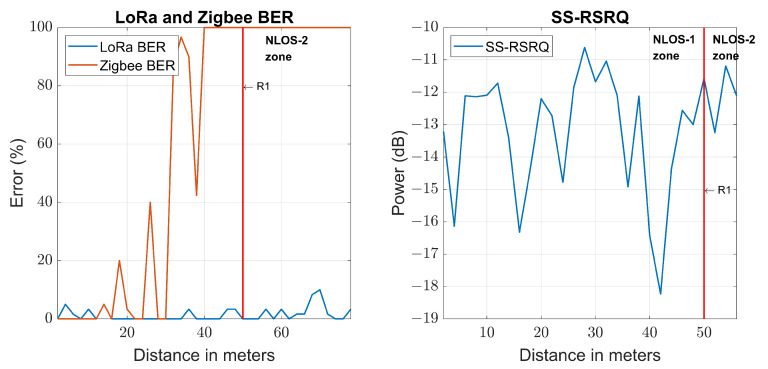
LoRa BER, Zigbee BER and 5G SS-RSRQ.

**Figure 15 sensors-23-03283-f015:**
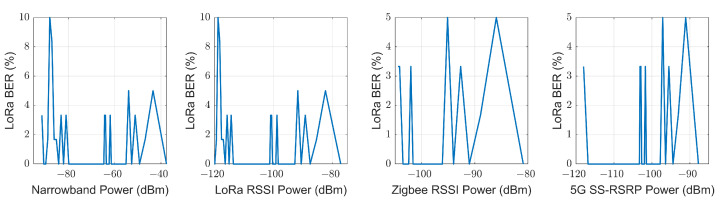
Comparison of empirical LoRa BER with the estimated received power for each technology.

**Figure 16 sensors-23-03283-f016:**
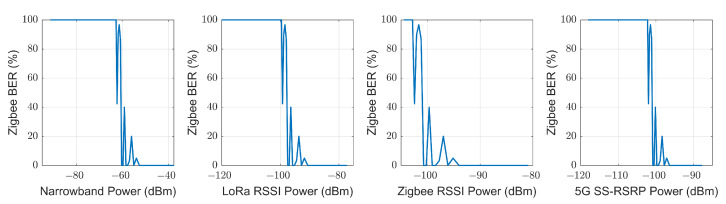
Comparison of empirical Zigbee BER with the estimated received power for each technology.

**Figure 17 sensors-23-03283-f017:**
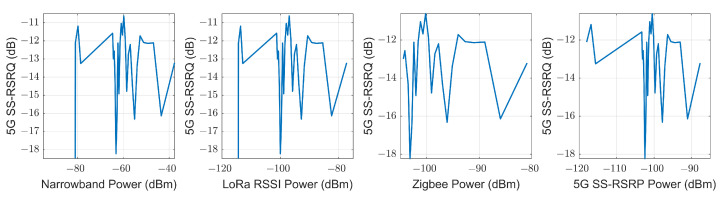
Comparison of empirical 5G SS-RSRQ with the estimated received power for each technology.

**Table 1 sensors-23-03283-t001:** Path loss measured in different scenarios.

Ref.	Modulation/Transmission Technology	Frequency	Model	n (Slope)
[16]	BPSK	868 MHz	CI	NSC 1.98, WSC 1.44
[26]	LoRa	868 MHz	CI	LOS 2.5–2.75
[28]	LoRa	920 MHz	CI	LOS 2.19, NLOS 3.87
[29]	LoRa	915 MHz	CI	LOS 3.03, NLOS 4.81
[31]	Zigbee	2.4 GHz	802.15.4A	LOS 1.63, NLOS 3.07
[35]	VNA	26 GHz	CI	Corridor: LOS 1.61, NLOS 3.80
[35]	VNA	26 GHz	CI	Stairwell: LOS 1.68, NLOS 4.18
[35]	VNA	38 GHz	CI	Corridor: LOS 1.69, NLOS 3.54
[35]	VNA	38 GHz	CI	Stairwell: LOS 2.76, NLOS 3.97
[36]	mBECS	28 GHz	CI	LOS 2.16, NLOS 3.81
[39]	CW	40 GHz	CI	LOS 1.8, NLOS 2.9
[39]	CW	40 GHz	FI	LOS 1.8, NLOS 2.9
[40]	VSG	3.5 GHz	FI	LOS 1.55, NLOS 2.96
[42]	V2X	5.89 GHz	CI	LOS 1.917
[43]	CW	1300 MHz	CI	LOS 1.5–1.8, NLOS 2.4–2.8
[44]	CW	1300 MHz	CI	LOS 1.79, NLOS 2.81
[45]	-	1300 MHz	CI	LOS 2.2
[46]	NB CW	914 MHz	CI	LOS 1.8–2.2, NLOS 3.25
[47]	CW	914 MHz	CI	LOS 1.9, NLOS 2.4
[48]	VNA	5 GHz	CI	LOS 1.7, NLOS 3.5
[49]	DPSK	850 MHz	CI	NLOS 3
[50]	SIMO	2.4 GHz	CI	LOS 2.1, NLOS 2.959
[51]	NB CW	2 GHz	CI	LOS 2.0, NLOS 2.8
[51]	NB CW	2.9 GHz	CI	LOS 1.6, NLOS 3.1
[52]	NB CW	4.5 GHz	CI	LOS 2.31, NLOS 3.69
[52]	NB CW	4.5 GHz	FI	LOS 1.32, NLOS 4.85

**Table 2 sensors-23-03283-t002:** SA configuration settings.

Parameters	Value
Center Frequency	868 MHz
SPAN	200 kHz
Frequency STEP	100 Hz
RBW	10 kHz
VBW	1 kHz
VBW Type	Linear
RBW:VBW	3
SPAN:RBW	100
Trace Type	Average
Detector Type	RMS/Avg
AVERAGES	100
SWEEP POINTS	100
Integration BW	40 kHz

**Table 3 sensors-23-03283-t003:** CI and FI models’ parameters for the two scenarios at 868 MHz.

Model	Transmission	Zone	$L_{0}$ (dB)	*n*	σ (dB)
CI	Narrowband	NLOS-1	31.21	1.97	5.08
		NLOS-2	31.21	3.03	4.56
CI	LoRa RSSI	NLOS-1	31.21	1.96	4.47
		NLOS-2	31.21	3.01	2.33
CI	Zigbee RSSI	NLOS-1	31.21	1.98	3.31
		NLOS-2	-	-	-
CI	5G SS-RSRP	NLOS-1	31.21	1.91	4.76
		NLOS-2	31.21	3.15	0.01
FI	Narrowband	NLOS-1	31.79	1.94	5.1
		NLOS-2	−45.94	7.26	3.89
FI	LoRa RSSI	NLOS-1	34.63	1.72	4.38
		NLOS-2	2.44	4.59	2.16
FI	Zigbee RSSI	NLOS-1	35.32	1.69	3.13
		NLOS-2	-	-	-
FI	5G SS-RSRP	NLOS-1	42.56	1.11	3.78
		NLOS-2	−42.65	7.39	0.013

**Table 4 sensors-23-03283-t004:** BER predictions from measurement.

	NB Power	LoRa RSSI	Zigbee RSSI	SS-RSRP
**LoRa BER (>5%)**	−88 dBm	−119 dBm	Out of range	Out of range
**Zigbee BER (>5%)**	−56 dBm	−94 dBm	−97 dBm	−98 dBm
**5G SS-RSRQ (<−18 dB)**	−81 dBm	−114 dBm	Out of range	Out of range

**Table 5 sensors-23-03283-t005:** BER for examples of received power values for each technology.

	Power Example	LoRa BER	Zigbee BER	5G SS-RSRQ
**Narrow band power**	−61 dBm	0%	86.93%	−11.04 dB
**LoRa RSSI**	−98 dBm	0%	86.67%	−11.04 dB
**Zigbee RSSI**	−91 dBm	0%	0%	−12.14 dB
**5G SS-RSRP**	−103 dBm	3.33%	100%	−13 dB

## Data Availability

The data presented in this study are available on request from the corresponding author.

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
