# Peer review of "LoRa, Zigbee and 5G Propagation and Transmission Performance in an Indoor Environment at 868 MHz"

_sensors, 2023, doi:10.3390/s23063283_

Round 1
Reviewer 1 Report
Improve quality of figures
Add some recent works related to the research
Author Response
We would like to thank you for your comments. We feel that the modifications made to the paper have truly contributed to its improvement. In order to take account of your comments, the following changes (which appear in red in the manuscript) have been made to the paper:
Improve quality of figures.
The authors thank the reviewer for the suggestion. All figures have been replaced by higher resolution versions.
Add some recent works related to the research:
The authors agree with the reviewer and, in this sense, we have added the following paragraph to the manuscript:
Similarly, the paper [52] presents results of the FI, CI and ABG models in an indoor environment for LOS and NLOS situations. In addition, it proposes a new model that improves the fit of the FI, CI and ABG models. Likewise, papers [53], [54], [55] and [56] present another approach to path loss model design based on the use of machine learning methods.
52. Majed, M.B., Rahman, T.A., Aziz, O.A., Hindia, M.N., and Hanafi, E. Channel characterization and path loss modeling in indoor environment at 4.5, 28, and 38 GHz for 5G cellular networks. International Journal of Antennas and Propagation 2018, 14
53. Levie, R., Yapar, Ç., Kutyniok, G., and Caire, G. Pathloss Prediction using Deep Learning with Applications to Cellular. Optimization and Efficient D2D Link Scheduling. Presented at the ICASSP 2020 - 2020 IEEE International Conference on Acoustics, Speech and Signal Processing (ICASSP), Barcelona, Spain, 2020; pp. 8678-8682.
54. Mao, K., Zhu, Q., Song, M., Li, H., Ning, B., Pedersen, G., and Fan,W. Machine-Learning-Based 3-D Channel Modeling for U2V mmWave Communications. IEEE Internet of Things Journal 2022, 9(18), 17592-17607.
55. Li, H., Chen, X., Mao, K., Zhu, Q., Qiu, Y., Ye, X., Zhong, W., and Lin, Z. Air-to-ground path loss prediction using ray tracing and measurement data jointly driven DNN. Computer Communications 2022, 196, 268-276.
56. Gui, J., Dai, X., and Deng, X. Stabilizing Transmission Capacity in Millimeter Wave Links by Q-Learning-Based Scheme. Mobile Information Systems 2020.
Reviewer 2 Report
This paper introduces the propagation and performance aspect of four transmissions under LOS and NLOS conditions in an indoor environment. Moreover, after measurements and proposing the path loss modeling, it also analyses the relations between the measured BER and the path loss models. The topic is interesting, but some comments need to be clarified as follows.
1. The references cited in this paper are old, and the format of the references is chaotic. It is recommended to quote more new studies and proofread the format specifications. Moreover, the overview of path loss prediction models are not sufficient. There have been some studies about path loss prediction models based on machine learning methods.
[1] Levie, R., Yapar, Ç., Kutyniok, G., & Caire, G. Pathloss prediction using deep learning with applications to cellular optimization and efficient D2D link scheduling. ICASSP’20, May, 2020, pp.8678-8682.
[2] K. Mao, Q. Zhu, M. Song, H. Li, B. Ning, G. F. Pedersen, et al., Machine learning-based 3D channel modeling for U2V mmwave communications, IEEE Internet Things J., Sept. 2022, vol. 9, no. 18, pp. 17592-17607.
[3] Li, H., Chen, X., Mao, K., Zhu, Q., Qiu, Y., Ye, X., ... & Lin, Z. Air-to-ground path loss prediction using ray tracing and measurement data jointly driven DNN. Computer Communications, 2022, 196, 268-276.
2. In the description of Figure 2, “The measurement points located in the main corridor, a horizontal corridor, are what we consider to be LOS measurements”. However, the TX position marked in Figure 2 can’t realize the measurement purpose of “The measurement points located in the horizontal corridor are considered to be LOS measurements”. Please check whether the TX position is correct, or give a reasonable explanation.
Moreover, it is suggested that the measurement positions of LOS and NLOS in Figure 2 should be marked by different colors for easily understanding.
3. In Figure 12, the titles of two subfigures are wrong, where “LOS” should be “NLOS”.
4. The spacing between cells in Table 2 is too large.
5. The descriptions about three link quality parameters are not clear. It is suggested to add more details.
6. It is suggested to move Figure 10 to the back of Sec.5.1.4.
Author Response
We would like to thank you for your comments. We feel that the modifications made to the paper have truly contributed to its improvement. In order to take account of your comments, the following changes (which appear in red in the manuscript) have been made to the paper:
1. The references cited in this paper are old, and the format of the references is chaotic. It is recommended to quote more new studies and proofread the format specifications. Moreover, the overview of path loss prediction models are not sufficient. There have been some studies about path loss prediction models based on machine learning methods.
[1] Levie, R., Yapar, Ç., Kutyniok, G., & Caire, G. Pathloss prediction using deep learning with applications to cellular optimization and efficient D2D link scheduling. ICASSP’20, May, 2020, pp.8678-8682.
[2] K. Mao, Q. Zhu, M. Song, H. Li, B. Ning, G. F. Pedersen, et al., Machine learning-based 3D channel modeling for U2V mmwave communications, IEEE Internet Things J., Sept. 2022, vol. 9, no. 18, pp. 17592-17607.
[3] Li, H., Chen, X., Mao, K., Zhu, Q., Qiu, Y., Ye, X., ... & Lin, Z. Air-to-ground path loss prediction using ray tracing and measurement data jointly driven DNN. Computer Communications, 2022, 196, 268-276.
The authors agree with the reviewer and, in this sense, we have added the following paragraph to the manuscript:
Similarly, the paper [52] presents results of the FI, CI and ABG models in an indoor environment for LOS and NLOS situations. In addition, it proposes a new model that improves the fit of the FI, CI and ABG models. Likewise, papers [53], [54], [55] and [56] present another approach to path loss model design based on the use of machine learning methods.
All references have been modified to have the ACS Style citation.
2. In the description of Figure 2, “The measurement points located in the main corridor, a horizontal corridor, are what we consider to be LOS measurements”. However, the TX position marked in Figure 2 can’t realize the measurement purpose of “The measurement points located in the horizontal corridor are considered to be LOS measurements”. Please check whether the TX position is correct, or give a reasonable explanation.
Thanks for the reviews, the authors have decided to change the zone named LOS to a zone named NLOS-1, and the NLOS zone to NLOS-2. In addition, changes have been made in the manuscript, appearing in red, to provide consistency to this change. In this way we consider the condition of the main corridor as NLOS. Therefore, the TX is correctly placed in Figure 2, since the objective of placing it inside a laboratory was to force two NLOS situations to considerably decrease the power of the transmission in order to be able to obtain errors in the LoRa transmission, because LoRa is a technology with long range and resistance to errors.
The color of the NLOS-2 zone in Figure 2 has been changed.
3. In Figure 12, the titles of two subfigures are wrong, where “LOS” should be “NLOS”.
The authors thank the reviewer for the suggestion. The error in figure 12 has been corrected.
4. The spacing between cells in Table 2 is too large.
The authors thank the reviewer for the suggestion. Table 2 has been corrected.
5. The descriptions about three link quality parameters are not clear. It is suggested to add more details.
The authors agree with the reviewer and, in this sense, we have added the following paragraph to the manuscript:
Quality Metric:
The quality parameter used for LoRa and Zigbee transmissions is the BER measured by the transceiver modules. BER is a parameter that represents the number of bit errors divided by the total number of bits received in a transmission. For 5G transmission, the SS-RSRQ parameter measured by the SA is used as the quality parameter, because it is used in 5G NR networks to determine the radio channel quality.
6. It is suggested to move Figure 10 to the back of Sec.5.1.4.
The authors thank the reviewer for the suggestion. Figure 10 has been moved to the back of section 5.1.4.
Reviewer 3 Report
The article "LoRa, Zigbee and 5G Propagation and Transmission Performance in an Indoor Environment at 868 MHz" is showing some merit in the field of study. Need minor revision to process to next level.
1) The study at different environmental conditions needs to be analyzed instead of closed environment
2) Is this model can be applicable for testing Sub 6 GHz applications and the futuristic needs of 6G.
3) Comparative analysis with literature in a tabular form before the conclusion will add novelty of your work in application perspective
4) Signal analyzer will provide additional parameters rather than spectrum analyzer. if possible try to extract additional parameters.
Author Response
We would like to thank you for your comments. We feel that the modifications made to the paper have truly contributed to its improvement. In order to take account of your comments, the following changes (which appear in red in the manuscript) have been made to the paper:
1) The study at different environmental conditions needs to be analyzed instead of closed environment
The authors thank the reviewer for the suggestion. Nevertheless, the aim of our work is to evaluate the performance, BER, of different technologies and relate them to the RSSI. Since we use LoRA, with a high dynamic range, we had to design an environment where a high path loss occurred within a reasonable distance. Therefore, the measurements were made in a controlled indoor environment, where we could observe errors for the selected technologies. In that case, we could relate firstly the relationship between real path loss and RSSI for each of them, and later to the QoS parameter.
2) Is this model can be applicable for testing Sub 6 GHz applications and the futuristic needs of 6G.
The model is applicable for LoRa, Zigbee and 5G. It will be applicable for other technologies that are similar in transmission technology and frequency.
Furthermore, we have compared three technologies with different frequency bands and modulations. We think that future 6G systems working in the same frequency range will also follow the same trend.
3) Comparative analysis with literature in a tabular form before the conclusion will add novelty of your work in application perspective.
Thanks for your suggestion. At the end of section 5.1.5 (Discussion of Path Loss Models) we outline several conclusions about the results of the models in this paper. These conclusions already comment on the results of the literature table, highlighting observed results such as:
In general, for LOS we obtain slope values close to 2. While for NLOS the slope values are close to 3. Which are very similar values to the results of our work.
4) Signal analyzer will provide additional parameters rather than spectrum analyzer. if possible try to extract additional parameters.
The authors thank the reviewer for the suggestion. We have used a 32 GHz spectrum analyzer Anritsu MS2090A. In the case of 5G, we have the 5GNR option to analyze 5G signals, so we can measure the S-RSRP, SS-RSRQ and SS-RINR parameters.
In the case of Zigbe and LoRA, these additional parameters were implemented by programming a Pycom Lopy and XBee transponder, respectively. Using these transponders, we get RSSI and BER by sending packets. With this, we are able to relate the loss in Path Loss, RSSI and BER according to a given position in our environment.
Round 2
Reviewer 2 Report
We have no further comments.